# Application of the Model of Spots for Inverse Problems

**DOI:** 10.3390/s23031247

**Published:** 2023-01-21

**Authors:** Nikolai A. Simonov

**Affiliations:** Valiev Institute of Physics and Technology of Russian Academy of Sciences, Moscow 117218, Russia; nsimonov@ftian.ru; Tel.: +7-925-348-1997

**Keywords:** inverse problems, image reconstruction, vague figures, mental images, artificial intelligence

## Abstract

This article proposes the application of a new mathematical model of *spots* for solving inverse problems using a learning method, which is similar to using deep learning. In general, the spots represent vague figures in abstract “information spaces” or crisp figures with a lack of information about their shapes. However, crisp figures are regarded as a special and limiting case of spots. A basic mathematical apparatus, based on L4 numbers, has been developed for the representation and processing of qualitative information of elementary spatial relations between spots. Moreover, we defined L4 vectors, L4 matrices, and mathematical operations on them. The developed apparatus can be used in Artificial Intelligence, in particular, for knowledge representation and for modeling qualitative reasoning and learning. Another application area is the solution of inverse problems by learning. For example, this can be applied to image reconstruction using ultrasound, X-ray, magnetic resonance, or radar scan data. The introduced apparatus was verified by solving problems of reconstruction of images, utilizing only qualitative data of its elementary relations with some scanning figures. This article also demonstrates the application of a spot-based inverse Radon algorithm for binary image reconstruction. In both cases, the spot-based algorithms have demonstrated an effective denoising property.

## 1. Introduction

Imaging based on the scan data of the object under study, and the processing of scattered signals received by sensors, refers to inverse problems. Relevant direct problems are the modeling of wave signals scattered from an object with a known distribution of material properties within it. [1,2]. In particular, medical devices that provide such imaging are CT, MRI, microwave tomography, electrical resistance and capacitance tomography, ultrasound imaging, and others [3]. There are other areas of application for such image reconstruction, including radar, ground-penetrating radar, and through-the-wall radar. Geophysics also uses visualization obtained by sounding the earth with the help of sound or electrical impulses, etc. [1,4]. Imaging in all these areas, with the exception of MRI, is associated with the solution of *inverse scattering* problems in one or another approximation [5].

The fundamental point here is that practically it is impossible to obtain a mathematically exact solution to the inverse problem, however, it is possible to approximately reconstruct an image with a finite spatial resolution. Generally speaking, inverse problems relate to ill-posed mathematical problems, and such property can be explained by a lack of information for an exact solution due to the noise and the finite amount of sensor signals. Therefore, approximate solution methods are used that utilize regularization, filtering, interpolation, and other approaches [3]. For example, this applies to CT, MRI, and ultrasound, as well as to studies on microwave tomography [6,7,8,9,10].

Note that conventional approximate reconstruction methods such as filtered back-projection in CT or simple inverse FFT in MRI are not always adequate and may lead to artifacts. Therefore, new methods based on appropriate models have been developed that more strictly take into account the physics of objects and the real properties of sensors [11,12,13,14]. In a rigorous formulation, the inverse problem is considered a nonlinear optimization with the regularization to find the minimum of the residual error (the norm of deviation of the received and calculated sensor data). Its solution is sought by the iterative method, where the direct problem is solved and the current residual error is calculated at each iteration step [5]. This rigorous approach is especially relevant for microwave tomography, where one has to solve the nonlinear inverse problem of electromagnetic scattering. Unfortunately, using simple and approximate reconstruction methods is inadequate here, since microwave scanning of a part of the body is performed in the near-field area of antennas; furthermore, multi-pass scattering effects are significant [6,7,8,9,10]. A big issue is that iterative solutions, especially for electromagnetic scattering, require a long execution time and consume large computer resources [7].

In recent years, much attention has been paid to solving nonlinear inverse problems using the artificial intelligence (AI) approach, applying neural networks of deep learning [15,16,17,18,19,20,21]. The idea of this method is that if the neural network is successfully trained on examples, then when new scattering signals are fed to the input of the network, the trained system can create the desired image directly on the basis of the acquired knowledge, without solving the complex inverse scattering problem. Another area of application of learning neural networks is their use for image recognition, classification, and segmentation [22,23,24].

However, many authors point out the disadvantage of the traditional model of artificial neurons in neural networks, which consists of excessive computational accuracy and an excessive number of adjustable system parameters [25]. Indeed, if the apparatus of 32-bit floating-point numbers are utilized in learning algorithms for input signals, weight coefficients, activation, and loss functions, then a lot of the training operations lead to large consumption of computer resources and a long execution time. For example, the training time for image recognition (which is a simple task for humans) can be on the order of several weeks even in high-performance systems [26]. It cannot be implemented on compact devices with limited resources and limited power consumption.

Therefore, intensive research is currently underway to develop algorithms using reduced bit width of numbers: 8 bits, 4 bits, 2 bits, and even 1 bit. This provides a rough approximation for all the network parameters and such optimized neural networks are called binarized [25]. It should be noted that deep learning networks, which are used, for example, for image recognition tasks, usually operate with 32-bit numbers, while it is obvious that high accuracy is not required for such an image classification [26].

A new mathematical model of spots is proposed in [27,28] for the representation and processing of incomplete, inaccurate, or qualitative information. In particular, it allows for the creation of algorithms of a new type for AI and building neuromorphic systems, which operate in a way that is close to human thinking. Instead of real numbers, the proposed model introduces logical L4 numbers, L4 vectors, and L4 matrices. The spot model allows one to represent mental images and semantic content of information, as well as make classification and fragmentation. A new approach to machine learning has also been suggested, which is applicable for solving inverse problems. In addition, a new architecture of neural networks is proposed, consisting of layers that are modeled by L4 matrices, input and output data are L4 vectors, and weight coefficients are L4 numbers [27].

Thus, in the spots model, information representation, storage, and processing are carried out using 4-bit logical L4 numbers, rather than 32-bit numbers that can significantly reduce the consumption of machine resources. Behind this, the developed learning algorithm does not use such complex calculations and iterations as in the backpropagation algorithm for learning from examples [29].

The main idea of the spot theory is that the synthesis of a large amount of even insignificant information allows one to extract detailed and even quantitative information about the object of interest. Since the ill-posedness of inverse problems leads to indefinite, fuzzy, and ambiguous solutions, the application of the spot apparatus seems to be quite adequate.

The proposed model turned out to be ideologically close to the research areas of mereotopology and qualitative geometry [30,31,32,33,34,35,36,37,38,39,40,41,42,43,44,45,46,47], the idea of which was laid down by Whitehead in 1929 [30]. On the other hand, the basic ideas of the spots model are also close to the rough set theory [48,49,50,51,52,53,54,55], the formal concept analysis [52,56,57,58], and the fuzzy set theory [59], including fuzzy geometry [60,61]. Moreover, the suggested concept is in good agreement with the ideology of granular computing [62,63,64,65,66,67,68].

Instead of points, mereotopology uses regions of space as primitive spatial entities and utilizes qualitative information about their relations. Among other areas, it has been applied to geographical information science, and image analysis [38]. One of the important fields of mereotopology is region connection calculus (RCC) [34,35,36], which has two variants. RCC-8 defines eight relations between regions, including overlapping, disconnection, external connection, and connections (touch) of the region’s boundaries. Bennett [42], as well as Jonsson and Drakengren [43] considered a shortened version of these relations—RCC-5, which does not consider the boundaries connection. A feature of RCC-5 is the uncertainty of boundaries, since here it is impossible to distinguish internal points from boundary ones.

Although most authors considered spatial relations as logical values, Egenhofer et al. [44,45] encoded them in form of logical tables. Namely, they introduced the concepts of 4-intersection [44] and 9-intersection [45] matrixes, which are logical matrices that encode the spatial relations between spatial regions. Notice that these matrixes are similar but differ from the L4 numbers proposed in [27,28] because authors of [44,45] also consider relations with the boundaries. Clementini et al. [46] generalized 9-intersection matrixes, replacing intersections for the crisp boundary with intersections for broad boundaries. Stell [47] also considered the way of representation for spatial relations using 3×1 logical vectors created on the base of notions part and compliment only. Finally, Butenkov et al. [69] introduced 2×2 logic tables for Cartesian granules, which are equivalent to L4 numbers for spots, and applied them for spatial data mining algorithms.

The rough set theory suggested by Zdzisław Pawlak [48,50] is a mathematical approach to the representation of the vagueness. One of the main advantages of the rough set approach is the fact that it does not need any preliminary or additional information about data, such as probability in statistics, or membership in the fuzzy set theory. This theory regards sets with incomplete information that does not allow to distinguish elements in some of their subsets, which are called granules. Pawlak’s theory introduced such notions as the lower and upper approximations of rough sets, the boundary region, and even the membership function for elements, which is similar to that for the fuzzy sets.

A general formulation and consideration of granules, including the problem of information granulation, which was later called the concept of granular computing, were first carried out by Zadeh in [62]. His definition of granules: “the information may be said to be granular in the sense that the data points within a granule have to be dealt with as a whole rather than individually” corresponds to the equivalence classes of the universe. Zadeh regards both crisp and fuzzy granules and “considers granular computing as a basis for computing with words” [63]. The elements of a granule are indiscernible that “depends on available knowledge” [65]. The importance of the application of granulation and granular computing relates to the fact that such approximation can lead to simplification in solving practical problems.

The graph is a mathematical model convenient for the representation of the structure of links (labeled edges) between elements (nodes or vertexes) of the system under study [70]. Nowadays, the apparatus of graphs is well suited for the analysis and processing of digital images in digital geometry [71], and the analysis and metrics of the structure of the physical connection of brain neurons [72]. On the other hand, graph theory is actively used in AI to model semantic networks in the knowledge bases, which are called knowledge graphs [73,74,75,76,77]. Note that, unlike spots, the graph is only an abstraction for the representation of a structure of the relations between the entities, rather than a spatial object. However, recent works utilize graphs embedded in some continuous space to reduce the dimension of the graph when processing its data [75,76,77].

Despite the ideological closeness to these theories, the proposed model of spots has a significantly different nature, since spots are not based on sets or fuzzy set concepts, and spot elements do not define. Instead of the elements, a spot can have a structure inside that is determined from spatial relations with other spots. Having elementary spatial properties, spots combine the concepts of discreteness and continuity, while graphs are discrete mathematical objects. Generally, the presence of similar mathematical models allows us to use some approaches and ideas from them.

## 2. Definition of Spots and the Apparatus of L4 Numbers

Spots are mathematical objects with elementary spatial properties, for which their inner region, outer region (environment), and a logical connection between these regions for any two spots are defined. The logical connection ab of two spots a, b is determined by two axioms [31,35]
(1)∀a, aa=1 (logical)
(2)∀a∀b, ab=ba

Environments $\tilde{a},\tilde{\;b}$ of spots a, b are also considered to be spots, therefore, a logical connection is also defined for them, satisfying the axioms (1) and (2). Axiomatically, we regard spots do not connect their environments, that is
(3)$$a\tilde{a}=0,\;b\tilde{b}=0$$


In general, the “shape” of spots and the properties of their environment, such as dimension and curvature of space, are not predetermined but can be evaluated from qualitative information about their elementary spatial relations (ER) to other spots, such as separation, intersection, inclusion, indistinguishability, etc. We consider crisp geometric figures as a special limiting case of spots.

Note that the connection can be defined not only in the case of the existence of a common region of space between two spots but also in a more general sense. For example, two geometric figures can be considered indistinguishable if they can be precisely coincided by a rigid movement. In general, any spot mapping can be defined with help of ER.

### 2.1. Definition of L4 Numbers

The elementary relations can be formalized using logical L4 numbers [27]. For spots a, b and their environments $\tilde{a}$, $\tilde{b}$ the L4 number 〈a|b〉 is defined as a table
(4)$$\langle a|b\rangle =\left[\begin{array}{cc} ab & a\tilde{b}\\ \tilde{a}b & \tilde{a}\tilde{b} \end{array}\right]$$
where ab, $a\tilde{b}$… denote the logical connections. Such L4 numbers, in general, permit distinguishing up to 16 different ER between spots. Examples of the ER and their corresponding L4 numbers are shown in Table 1.

We call these spatial relations as elementary relations because they carry the lowest-level qualitative information about spots. However, a large amount of such qualitative data allows for extracting higher-level qualitative information and even numerical information.

The mathematical apparatus of the spot model is based on the L4 numbers, rather than on real numbers. As far as the basis of this apparatus is described in more detail in the previous works [27,28], here we will briefly outline the main content and reveal the meanings of the concepts introduced there.

In [27,28], the basis of spots is defined as a collection of “known” spots that can be in some mutual ER. The representation of a spot by their ER on the basis of spots is a mapping or imaging of the spot on this basis. Note that the system of basis spots is analogous to the numerical basis functions, and the orthogonality of the base functions is analogous to the separated basis spots, which we call *orthogonal* spots. Let us call the collection of spots with a certain ER between them the structure of spots. The structure of the basis spots included in a spot a will also be called the *structure* of the spot a.

Note that spot mapping is generally similar to the concept of projection for crisp figures. Consequently, the spot is analogous to some volumetric object, which is determined by its projections on different planes. Hence, one can improve knowledge about the structure of the spot by fusion its mappings on different bases into a “volumetric” image.

Let us define the operations union ∨ and the intersection ∧ for the spots, which permits the creation of new spots. We suggest the following definitions, which are similar but different from those of the set theory:(5)$$\begin{array}{c} c=a\vee b\;↔\;\forall x\;\left(cx=ax+bx\right)\\ d=a\wedge b\;↔\;\forall x\;\left(\tilde{d}x=\tilde{a}x+\tilde{b}x\right) \end{array}$$

Here, the symbol + denotes the logical disjunction operation. Note that, in contrast to the sets, (5) does not define the images of spots c, d absolutely, because they depend on the spots’ basis $\left\{x_{i}\right\}$. Following the equality $c\tilde{c}=0$, see (3), it is possible, for example, to derive the following equations from (5):(6)$$c=\underset{i}{\vee }x_{i},\;x_{i}\;:\left(\tilde{a}x_{i}+\tilde{b}x_{i}=0\right)$$

The definitions (5) also permit to derive simple properties for zero spots ∅:(7)a∨∅=a, a∧∅=∅
and express intersection parts A, B, C, and D of spots a and b in Figure 1, using the operation ∧:(8)$$A=a\wedge \tilde{b},\;B=\tilde{a}\wedge b,\;C=a\wedge b,\;D=\tilde{a}\wedge \tilde{b}$$

### 2.2. Definition and Geometric Meaning of L4 Vectors and L4 Matrices

As mentioned earlier, information about any spot can be defined by its mapping on some basis. Then it can be encoded as a vector with coordinates, corresponding to its ER with basis spots. Such a vector of L4 numbers is called an L4 vector. For example, the L4 column vector $a_{X}$ [27] of the spot a, represented on the base $X=\left\{x_{i}\right\}$, is
(9)$$a_{X}\equiv \left[\langle a|x_{1}\rangle ;\;\langle a|x_{2}\rangle ;\ldots ;\;\langle a|x_{n}\rangle \right]$$
where *n* is the number of spots in the basis X. L4 vector (9) is similar to a numerical vector but its elements are L4 numbers. On the other hand, mapping (9) is also similar to the projection of a 3D body on some plane, which, can only represent partial information about the body.

Papers [27,28] introduce the idealized concept of *atomic* basis and atomic spots, which do not intersect each other and other spots. Note that the atomic spots are similar to points, pixels, voxels, or elements of sets. Another useful notion is *orthogonal* spots, for which their mutual ER is separated. For example, intersection parts of spots are orthogonal and can be regarded as some approximation for the atomic basis.

The L4 matrix A=〈Y|X〉 defines ER between the spots of two bases, $X=\left\{x_{i}\right\}$ and $Y=\left\{y_{j}\right\}$ and formalizes the mapping from X basis to Y basis:(10)$$\langle Y|X\rangle \equiv \left[\langle y_{j}|x_{i}\rangle \right]=\left[{\left(y_{1}\right)}_{X};\;{\left(y_{2}\right)}_{X};\ldots ;\;{\left(y_{n}\right)}_{X}\right]$$

Here, ${\left(y_{j}\right)}_{X}$ are row L4 vectors of spot $y_{j}$, represented on the basis X. Note that the L4 matrix can be used to transform the L4 vector from one to another basis [27] which is similar to the mapping function in topology and geometry. Formally, it can be represented in the form of
$$a_{Y}=\langle Y|X\rangle \;a_{X}$$
however, there is no simple solution to define the rules for such a product, and we will address this issue in the next section. The exception is the special case of the L4-matrix, which we call the indistinguishability matrix that is similar to the numerical identity matrix. The indiscernibility matrix I has diagonal elements corresponding to the indiscernibility and all other elements—to separation. Then multiplication of the L4 matrix I and any L4 vector a corresponds to an identity transformation:a=I a

There is a special case when all the spots of two bases are separated that is analogous to orthogonal coordinates in geometry. It is obvious that in this case, it is impossible to obtain a mapping transformation using the product of an L4 matrix and L4 vector, and it is necessary to obtain additional independent data.

### 2.3. Multiplication Rules for L4 Vectors and L4 Matrixes

First, consider the simplest case of an atomic basis $A=\left\{u_{i}\right\}$, where basis spots are orthogonal and do not intersect other spots [27,28]. For it, one can define an ER ${\langle a|b\rangle }_{A}$ between the spots a and b with respect to the basis A and the “scalar” product of vectors $\left(a_{A},\;b_{A}\right)$ according to the rule
(11)$${\langle a|b\rangle }_{A}=\left(a_{A},\;b_{A}\right)=\left[\begin{array}{cc} \overset{n}{\underset{i=1}{\sum }}au_{i}\cdot bu_{i} & \overset{n}{\underset{i=1}{\sum }}au_{i}\cdot \tilde{b}u_{i}\\ \overset{n}{\underset{i=1}{\sum }}\tilde{a}u_{i}\cdot bu_{i} & \overset{n}{\underset{i=1}{\sum }}\tilde{a}u_{i}\cdot \tilde{b}u_{i} \end{array}\right]$$
where the symbol “∙” denotes a logical conjunction. We will apply the same rule for an orthogonal basis $U=\left\{u_{i}\right\}$, consisting of separated spots.

Let us regard a spot a, basis $B=\left\{b_{i}\right\}$ and an atomic basis $A=\left\{u_{i}\right\}$. We suppose that a and all $b_{i}$ spots can be mapped on the atomic basis A. Then the rule for the product of the L4 matrix 〈B|A〉 (10) and L4 vector $a_{A}$ (9) can be defined as the following:(12)$$a_{B}=\langle B|A\rangle \;a_{A}=\left[{\langle a|b_{i}\rangle }_{A}\right]$$
where L4 number ${\langle a|b_{i}\rangle }_{A}$ is defined in (11). Note that Equation (12) defines the transformation of the mapping of the spot a from basis A to basis B.

For an arbitrary basis $X=\left\{x_{i}\right\}$, when spots $x_{i}$ can be intersected, the definition of ${\langle a|b\rangle }_{X}$ is more complicated. First, let us consider the orthogonal basis $U=\left\{u_{i}\right\}$ of all intersections of the spots in X and find the vectors $a_{U}=\left[\langle a|u_{k}\rangle \right]$ and $b_{U}=\left[\langle b|u_{k}\rangle \right]$ on the basis U. Then, we define the following equality for calculation ${\langle a|b\rangle }_{X}$:$${\langle a|b\rangle }_{X}=\left(a_{U},\;b_{U}\right)={\langle a|b\rangle }_{U}$$
and apply the rule (11). We define the vectors $a_{U}$ and $b_{U}$ using the following formal matrix equations:(13)$$a_{U}=\langle U|X\rangle \;a_{X},\;b_{U}=\langle U|X\rangle \;b_{X}$$
where 〈U|X〉 is the L4 matrix that consists on $\langle u_{i}|x_{j}\rangle$ elements and is used for mapping vectors from basis X to basis U.

To determine the transformation rule for (13), we first apply a convenient method of numbering the intersections $\left\{u_{k}\right\}$ of spots $\left\{x_{i}\right\}$, using a binary code. Namely, generalizing (8), each $u_{k}$ can be defined in terms of the spots $x_{i}$ or ${\tilde{x}}_{j}$ connected by operator ∧. For example, the binary index $k=101\ldots 0_{2}$ corresponds to the following spot $u_{k}$ [28]:(14)$$u_{k}=x_{1}\wedge {\tilde{x}}_{2}\wedge x_{3}\ldots \wedge {\tilde{x}}_{n}$$

ER $\langle a|u_{k}\rangle$ for any spot a and for $u_{k}$ (14) can be found using the following approximate equation [28]
(15)$$\langle a|u_{k}\rangle =\left[\begin{array}{cc} ax_{1}\cdot a{\tilde{x}}_{2}\cdot \ldots \cdot a{\tilde{x}}_{n} & a{\tilde{x}}_{1}+ax_{2}+\cdots +ax_{n}\\ \tilde{a}x_{1}\cdot \tilde{a}{\tilde{x}}_{2}\cdot \ldots \cdot \tilde{a}{\tilde{x}}_{n} & \tilde{a}{\tilde{x}}_{1}+\tilde{a}x_{2}+\cdots +\tilde{a}x_{n} \end{array}\right]$$
which defines the rule for the product $\langle U|X\rangle \;a_{X}$ in (13). A similar equation can be written for the spot b to determine the product $\langle U|X\rangle \;b_{X}$ in (13).

Equation (15) was tested in [28] when solving the problem of reconstruction of the shape of plane figures, processing its ER data with basis figures (Figures 4–6 in [28]). It turned out that (15) gives uncertainty in the form of a blurred boundary. However, it is possible to eliminate it if to apply additional rules, correcting ER $\langle a|u_{k}\rangle$ in (15):(16)$$\mathrm{if}\;\left\{\forall x_{j}:ax_{j}=0,\;u_{k}x_{j}=0\right\}\;then\;a>u_{k}\;\mathrm{if}\;\left\{\;\forall x_{j}:\tilde{a}x_{j}=0,\;u_{k}x_{j}=0\right\}\;then\;a<>u_{k}$$
where the symbols <> and > denote the separation and inclusion (more) relations, respectively (see Table 1).

Equations (15) and (16) help to determine the general rule for multiplying an arbitrary L4 vector $a_{X}$ and an L4 matrix A=〈Y|X〉 defined on the basis $X=\left\{x_{i}\right\}$ and $Y=\left\{y_{j}\right\}$. We can write this rule in the following form:(17)$$a_{Y}=\langle Y|X\rangle \;a_{X}=\langle Y|V\rangle \cdot \langle V|W\rangle \cdot \langle W|U\rangle \cdot \langle U|X\rangle \;a_{X}$$

Here the basis $U=\left\{u_{i}\right\}$ consists of the intersections of the points $x_{i}$, $V=\left\{v_{i}\right\}$ is the basis of the intersections of the spots $\left\{y_{j}\right\}$, and $W=\left\{w_{i}\right\}$ is the basis of the intersections of the spots of U and V basis. Note that Equation (17) should be considered as a series of transformations from one basis to another, namely
(18)$$a_{U}=\langle U|X\rangle a_{X},\;a_{W}=\langle W|U\rangle a_{U},\;a_{V}=\langle V|W\rangle a_{W},\;a_{Y}=\langle Y|V\rangle a_{V}$$

Product $\langle U|X\rangle \;a_{X}$ can be calculated, using (15) and (16), but the vectors $\langle V|W\rangle \;a_{W}$ and $\langle Y|V\rangle \;a_{V}$—using (11), (12), regarding V, W as an atomic basis. Finally, let us use the following natural rule for calculating the product $\langle W|U\rangle \;a_{U}$:(19)$$\mathrm{if}\;w_{k}<u_{i}\;\mathrm{then}\;\langle a|w_{k}\rangle =\langle a|u_{i}\rangle$$
where symbols < denotes relation inclusion (less) (see Table 1). Note that rule (19) is also approximate.

## 3. General Approach to Inverse Problems and Learning Using L4 Matrices

### 3.1. Solution of Inverse Problems

It follows from the definition of L4 matrices 〈Y|X〉 (10) that its inverse matrix 〈X|Y〉 is equal to
(20)$${\langle Y|X\rangle }^{-1}\equiv \langle X|Y\rangle =\left[\langle x_{i}|y_{j}\rangle \right]$$
and hence it must always exist and be equal to the transposed matrix 〈Y|X〉 with additional transposed elements (L4 numbers). Therefore, as it following from (17) and (20), formally the solution of the equation $a_{Y}=\langle Y|X\rangle \;a_{X}$ can be represented as
(21)$${\hat{a}}_{X}=\langle X|Y\rangle \;a_{Y}=\langle X|U\rangle \cdot \langle U|W\rangle \cdot \langle W|V\rangle \cdot \langle V|Y\rangle \;a_{Y}$$
where, as in (17), the basis U consists of intersections of the spots in X, the basis V—intersections of the spots in Y and W—intersections of the spots of U and V basis. Considering that Equations (15), (17), and (19) are approximate, we can conclude that in the general case, the inverse solution (21) is also approximate:$$a_{X}≅{\hat{a}}_{X}={\langle Y|X\rangle }^{-1}\;a_{Y}$$

### 3.2. Solving Inverse Problems Using L4 Matrices by Learning Method

As mentioned in the introduction, the practical application of solving inverse problems, especially electromagnetic inverse scattering, requires a large amount of computation time and resources. Alternative approaches involve the use of neural networks to train a solving system, which, after training, can make an inverse solution for newly measured data. In the spots model, this has an analogy with the situation when the matrix A=〈Y|X〉 in (17) is unknown and it is wanted to be determined on training examples. Let us evaluate an unknown L4 matrix A on the basis of a set of training examples $\left\{x_{i},\;y_{i}\right\}$, using the equality
(22)$$y_{i}=Ax_{i}$$

We can regard $x_{i},\;y_{i}$ as L4 vectors for spots $x_{i}$ and $y_{i}$ that form the basis $X=\left\{x_{i}\right\}$ and $Y=\left\{y_{i}\right\}$ of the training data. Then, we can compose an L4 matrix 〈Y|X〉 and represent the matrix A in (22) as:(23)$$A=\langle B_{Y}|Y\rangle \cdot \langle Y|X\rangle \cdot \langle X|B_{X}\rangle$$
where $B_{X}$ and $B_{Y}$ are atomic bases for L4 vectors $x_{i}$ and $y_{i}$, correspondingly. Obviously, for the testing set $\left\{x_{i},\;y_{i}\right\}$ the matrix 〈Y|X〉 is equal to the indistinguishability matrix I. Note that Equation (23) is a schematic interpretation of the learning process [29].

Let us consider the application (23) of the learning system to obtain the inverse solution of the following equation:
$$b=\langle B_{Y}|Y\rangle \cdot \langle Y|X\rangle \cdot \langle X|B_{X}\rangle \;a$$

To do this, we can use the transformation of the matrix A (23), similar to (21), to represent the inverse solution in the following general form:(24)$$a\;≅\;\hat{a}=\langle B_{X}|X\rangle \cdot \langle X|Y\rangle \cdot \langle Y|B_{Y}\rangle \;b$$

We should especially consider the case when input data c and/or output data d are numeric. Then, instead of matrixes $\langle Y|B_{Y}\rangle$ or $\langle B_{X}|X\rangle$ in (24) we have to apply the corresponding operators $B_{X}$ and $B_{Y}$ that transforms L4 data to numerical data or vice versa. Then, the forward problem has the form of
$$d=B_{Y}\left(c_{Y}\right)\cdot \langle Y|X\rangle \cdot B_{X}\left(c\right)$$
and its inverse solution instead of (24) can be represented in the following form:(25)$$c≅B_{X}^{-1}\left(d_{X}\right)\cdot \langle X|Y\rangle \cdot B_{Y}^{-1}\left(d\right)\;$$
where$\;B_{X}^{-1}$ and $B_{Y}^{-1}$ are the inverse operators.

### 3.3. Image Reconstruction by Processing Qualitative Data

Although the proposed theory is developed for spots, which in general correspond to vague figures, it is convenient to verify its mathematical apparatus on crisp figures, which are the limiting case of spots. Let us consider the figure under test as a conditionally unknown spot, and the figures, which are used for mapping this spot (or “sampling”) as the known basis of spots [28]. More specifically, we consider the shape reconstruction of a crisp plane figure, utilizing the only qualitative information of its ER with the bases figures without additional details about these relations. However, it is possible to infinitely refine the reconstructed shape of the unknown figure by increasing the number of samplings and processing all the ER data. It may seem surprising, but it is theoretically possible, to reconstruct the shape of an object with absolute precision. This is a consequence of the following theorem.

**Theorem** **1.**
*In order for two figures to be pointwise equal, it is necessary and sufficient that their elementary relations with any other figure of finite size be the same.*


**Proof.** **Necessity**. As can be seen from the diagram in Figure 2, the condition of equality of figures a and b is equivalent to the equality of both their intersection parts A and B to the zero figure ∅. Let us suppose there is a figure c that has different ER with a and b, i.e., different connection values with these figures. For example, ac=0, bc=1 (Figure 2). Then $\left\{\exists E=b\wedge c:aE=0\right\}\to E\subset B=\tilde{a}\wedge b\to \;B\neq \emptyset$. Therefore, a≠b, which proves the necessity condition.**Sufficiency**. Let us prove this by contradiction. Assume that for two figures a and b their ER is equal with any finite figure, but a≠b. Then, $\left(A\neq \emptyset \right)\;\vee \;\left(B\neq \emptyset \right)$ (see Figure 2). If, for example, B≠∅, then $\exists c:\left\{ac=0,\;Bc=1\right\}\to bc=1$ that contradicts the condition of equality ER with any finite figure. Therefore, the assumption a≠b is false, which proves the sufficiency condition. □

It follows from Theorem 1 that all information about the shape of each figure is contained in the infinite set of its ER with all other figures of finite size. Therefore, in principle, it is possible to reconstruct this shape using such qualitative information. However, due to the incomplete, finite amount of ER data, figure shape reconstruction can only be approximate. This corresponds to the fact that the result of such a reconstruction corresponds to a blurry figure, that is, to a spot.

Note that the shape’s reconstruction, by processing qualitative data, refers to inverse problems. Indeed, its forward problems can be formulated as
(26)$$a_{X}=\langle X|P\rangle \;a_{P}$$
where P is a basis for atomic spots—pixels or voxels, $X=\left\{x_{i}\right\}$ is a basis for scanning figures for testing, $a_{X}$—L4 vector of ER data for the reconstruction of the figure under test $a_{P}$. Following (21), the reconstructed figure ${\hat{a}}_{P}$ is the inverse solution of (26) that, similar to (21), can be represented in the form of the equation
(27)$${\hat{a}}_{P}=\langle P|X\rangle \;a_{X}=\langle P|U\rangle \cdot \langle U|X\rangle \;a_{X}$$
where U is the basis of intersections of spots $\left\{x_{i}\right\}$. The mapping $a_{U}=\langle U|X\rangle \;a_{X}$ can be found using (15) and (16).

### 3.4. Inverse Radon Algorithm for Binary Figures

Let us consider scanning figures–squares as the basis $X=\left\{x_{i}\right\}$**,** and use the calculated sinograms (projections) S of these squares as training data, which we will assign to the basis $Y=\left\{y_{i}\right\}$. As before, $\left\{x_{i},\;y_{i}\right\}$ will be considered training data for the learning system, and we will determine an algorithm for the inverse solution by learning.

The forward problem is the Radon transform Sec. 8.7.3 of [2] that can be written in the form of
(28)$$s=R\left(a_{P}\right)$$
where P is the basis of pixels, R is the Radon transform operator, s is the sinograms of the $a_{P}$ image. Following (25), the inverse Radon solution of (28) can be represented as
(29)$${\hat{a}}_{P}=\langle P|U\rangle \cdot \langle U|X\rangle \cdot \langle X|Y\rangle \cdot B_{Y}^{-1}\left(s\right)\;$$
where U is the basis of intersections of $\left\{x_{i}\right\}$. Note, the operator $B_{Y}^{-1}$ depends on the training sinograms data S and matrix 〈Y|X〉 is the indiscernibility matrix I for solving by learning methods, as it was mentioned before. Therefore,
(30)$${\hat{a}}_{P}=\langle P|U\rangle \cdot \langle U|X\rangle \;a_{X};\;a_{X}=a_{Y}=B_{Y}^{-1}\left(s\right)$$

Let us find rules for calculation $a_{Y}$ (30), defining such ER between sinograms that are presented in Figure 3. Here, small spiking sinograms (continue lines) correspond to relatively small basis squares $x_{i}$ and oval-type sinograms (dashed lines) correspond to ellipse figure $a_{P}$ (see Section 4.3).

The main idea of suggesting an algorithm is that if the basis figure $x_{i}$ has such ER with the figure under test a as separation or intersection, then there are projection angles for which their sinograms are separated or intersected. However, if $x_{i}$ is included in a then all their sinograms have ER inclusion as well. Hence, we can define the following rules for ER of projections $s_{X}\left(i,j,k\right)$ and $s_{a}\left(i,j,k\right)$, which are converted to *logical values*:(31)$$a_{X}\left(i\right)=\langle a|x_{i}\rangle =\left[\begin{array}{cc} ax_{i} & 1\\ \tilde{a}x_{i} & 1 \end{array}\right]\;ax_{i}=\sum_{k}\sum_{j}s_{a}\left(i,j,k\right)\cdot s_{X}\left(i,j,k\right)\;\tilde{a}x_{i}=\sum_{k}\sum_{j}\neg s_{a}\left(i,j,k\right)\cdot s_{X}\left(i,j,k\right)$$

Here, ¬ is the logical negation, i-index corresponds to that of $x_{i}$ square, j-index corresponds to the projection coordinate $\xi \left(j\right)$ and k-index corresponds to the projection angle $\alpha_{0}\left(k\right)$ Sec. 8.7.3 of [2]. Using (30) and (31) we get a spot-based inverse Radon algorithm for the reconstruction of binary images.

## 4. Results of Image Reconstruction

To illustrate the suggested theory, MATLAB programs were written that provide processing of ER data between the figure under test and basis spots (crisp figures) $x_{i}$, which are scanning (or basis) squares. To obtain a better resolution, we utilize quite tight distribution of the basis spots that makes the intersections $u_{k}$ (14) to be relatively small. We used a computer with an AMD Ryzen 7 2700 X processor, 8 cores, 3.70 GHz, 32GB RAM, and no GPU.

### 4.1. Reconstruction of Binary Images

The ER data were obtained using scanning of 4×4 pixels squares with the scan period 1 pixel and processed them using (27) and algorithm (15), (16). The number of basis squares was approx. 20,000, and the calculation time was about 9 min in all cases.

To compare the original and reconstructed binary images we calculated the misfit error mer
(32)$$mer=\frac{\left|N_{OI}-N_{RI}\right|}{N_{OI}}$$
where $N_{OI}$ and $N_{RI}$ are numbers of pixels that correspond to the inner regions of the original (noise-free) and the reconstructed images, correspondingly.

Figure 4 represents the reconstruction of images of a five-pointed star without and with strong noise, utilizing only data from its ER with scanning squares. Note that Figure 4d demonstrates the effective denoising capability of the algorithm (15), (16), (27). The image sizes were 128×128 pixels, and the misfit errors for the reconstructed images are mer = 0.1% for Figure 4b, and mer = 3.1% for Figure 4d.

Figure 5 demonstrates the reconstruction of hand-mask images—noise-free and strongly noisy, using similar ER data and rules. Note that Figure 5d also demonstrates the effective denoising capability of the algorithm (15), (16), (27). The image sizes were 120×120 pixels and the misfit errors for the reconstructed images are mer = 3.1% (for Figure 5b), and mer = 4.7% (for Figure 5d).

### 4.2. Reconstruction of Gray Scale Images

To be able to apply the developed reconstruction algorithm (15), (16), (27) to grayscale images, it is necessary to add a new dimension to 2D spots corresponding to their intensity value. In order for this numerical coordinate to be consistent with the general spot ideology, we represent the intensity axes as a linear structure, a chain of intersected spots (Figure 6).

For example, these spots can be numerical intervals, and hence we can split the grayscale image into flat layers, corresponding to these intervals. Then one can reconstruct images in the layers independently and combine them again into the entire image. Results of the reconstruction are demonstrated in Figure 7, Figure 8, Figure 9 and Figure 10, where the intensity axes of 128×128 pixels images were divided into 20 layers. The number of basis squares was about 20,000, and the calculation times were 6–9 min.

As before, the ER data were obtained using 4×4 pixels squares that were scanned in each of 20 layers, and their scan period was 1 pixel. The SNR, which is defined for an average image intensity, is 9.7 dB (for Figure 9c), 23.3 dB (for Figure 10c), and 9.3 dB (for Figure 10e). Note that Figure 9d and Figure 10d,f also demonstrate the noise reduction ability of the reconstruction algorithm (15), (16), (27).

### 4.3. Inverse Radon Image Reconstruction

We compared a conventional back-projection and a spot-based (31) algorithm for the reconstruction of binary images, which used under-sampled parallel-beam sinograms for 6, 9, and 18 projection angles only. Figure 11 and Figure 12 show the results of this comparison for the 128×128 pixels image reconstruction. The sinograms are calculated using the Radon transform, but they imitate the real experimental sinograms of X-ray transmission through the body in a CT system with parallel-beam geometry [3]. Note that typically a CT scanner collects projection signals in approximately 1° increments, and hence the simulated examples in Figure 11 and Figure 12 are indeed highly under-sampled.

Application of the back-projection algorithm with Hann filter is demonstrated in Figure 11 for two images of the ellipse: noise-free (Figure 11a) and strong noisy (Figure 11e). It is clear that the results of the noisy image reconstruction in Figure 11f–h demonstrate significant blurring for the reconstructed image.

Figure 12 shows the results of the same image reconstructions, using the spot-based algorithms (30) and (31) with 5×5 pixels square and 1 pixel scan period. The number of basis squares was about 20,000, and the calculation times were about 6 min. These results of the reconstruction demonstrate the fact that the suggested algorithm allows the reconstruction of unblurred images even for a small number of projection angles. Images in Figure 12f–h also demonstrate a strong denoising effect for the spot-based algorithm, in contrast to the filtered back-projection algorithm. The misfit errors mer (32) were 3.6% (for Figure 12b,f), 4.8% (for Figure 12c,g), and 5.6% (for Figure 12d,h).

## 5. Discussion

In [27], the use of the apparatus of the spots model for creating neural networks of a new type is considered, in which each layer corresponds to an L4 matrix and L4 numbers are used instead of real numbers. Here, the L4 vectors play the role of input and output signals for each layer, and the L4 matrix of each layer plays the role of the weight matrix. For example, Equations (21) and (23) can be implemented using such a neural network, which consists of 4 and 3 layers, respectively. In addition, it is possible to create a neural network in the form of a neuromorphic electronic device built on solid-state elements such as field-effect transistors (FETs), FeFETs, or memristors [27].

The potential advantage of the proposed neural networks over conventional ones is, in particular, that the L4 matrix apparatus does not use real numbers with complex calculations during iterations in the backpropagation algorithm of learning by examples. Instead, Equation (24) uses the inverse matrix product. Although the proposed algorithms are approximate, they are adequate to the fact that it is almost always impossible to obtain an exact solution to inverse problems due to the finite number of measured signals. In addition, the tasks of recognition and classification, in principle, do not belong to the class of tasks that require accurate calculations.

The reconstructed images in Figure 4, Figure 5, Figure 7, Figure 8, Figure 9, Figure 10, Figure 11, Figure 12 demonstrate a good denoising ability of the proposed algorithm. This property relates to the fact that scanning squares of 4×4 or 5×5 pixels plays the role of a spatial filter and averages the sampled data. However, the spatial resolution of the reconstructed image is determined by the scan period of 1 pixel. This can be explained using Formulas (15) and (27), from which it follows that the resolution corresponds to the intersections size of 1×1 pixels.

An imaging algorithm using the spot-based inverse Radon transformation (Equation (31) and Figure 12) illustrates the processing of qualitative data for solving by learning. Indeed, sampling figures are basis spots and also relates to the training set, whereas the figure under test corresponds to the test example in the machine learning paradigm [29]. Finally, the reconstructed image, which is mapped on the basis of the intersections, corresponds to the solution of the trained system.

As it was noted in the Introduction, we can draw a general conclusion about the ideological proximity of the models of the spots and rough sets [48,49,50,51,52], although they have a fundamental difference. In addition, there are several close concepts between the spot model and the rough set theory (see Table 2).

As it is shown in Table 2, spots are similar to granules, which is also the main concept in the granular computing (GC) research area [62,63,64,65,66,67,68]. A comparison of the spot and granule concepts in GC allows us to conclude that both models are also very close in many aspects.

The suggested spots model can be used for the theory of qualitative geometry (QG). For this application, it is necessary to introduce new concepts that are low-level analogies of the notion in the geometry and topology, including line, surface, dimensionality, curvature of space, etc. Based on the CG, one can introduce the concept of a semantic information space, which is an analogue of an information system characterized by an information table and is used, for example, in the theory of rough sets [50,52].

## 6. Conclusions

This article is devoted to the description of the concept and the basis of the apparatus of new mathematical objects–spots, which are introduced to represent and process qualitative data. It can be used to model human mental images, perceptions, and reasoning in AI. Furthermore, this paper demonstrates another application of the developed apparatus—for solving inverse problems by the learning method.

The proposed model uses such qualitative information about spots as elementary relations between them and introduces L4 logical numbers that encode these relations. Based on L4 numbers, the theory introduces L4 vectors and L4 matrices using the analogy with numerical matrix algebra. Although L4 numbers correspond to an elementary level of qualitative data, fusing a large number of them allows you to extract a higher level of information, including numerical.

Equations have been derived for reconstructing an image using only qualitative information about its elementary relations with a set of base figures. A general scheme for solving inverse problems for L4 and numerical data is proposed, including a learning method for solving.

The introduced apparatus was tested by solving image reconstruction problems using only qualitative data of its elementary relations with the scan figures. The application of spot-based inverse Radon’s algorithm for the reconstruction of a binary image was also demonstrated.

Further research in the field of the proposed theory involves the development of algorithms for solving various inverse problems, including inverse electromagnetic scattering. Another goal of the work is to design neural networks based on the proposed spots model, where each layer corresponds to the L4 matrix.

## Figures and Tables

**Figure 1 sensors-23-01247-f001:**
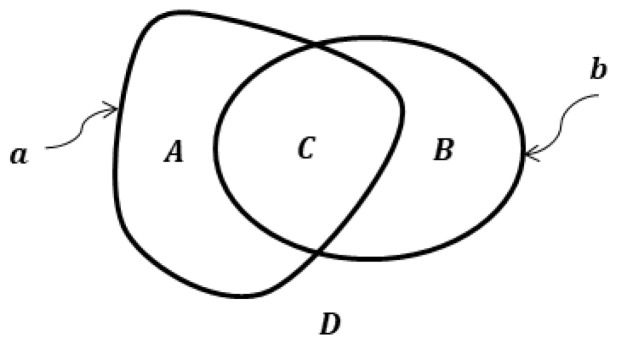
Euler-Venn diagram for the elementary relations between spots.

**Figure 2 sensors-23-01247-f002:**
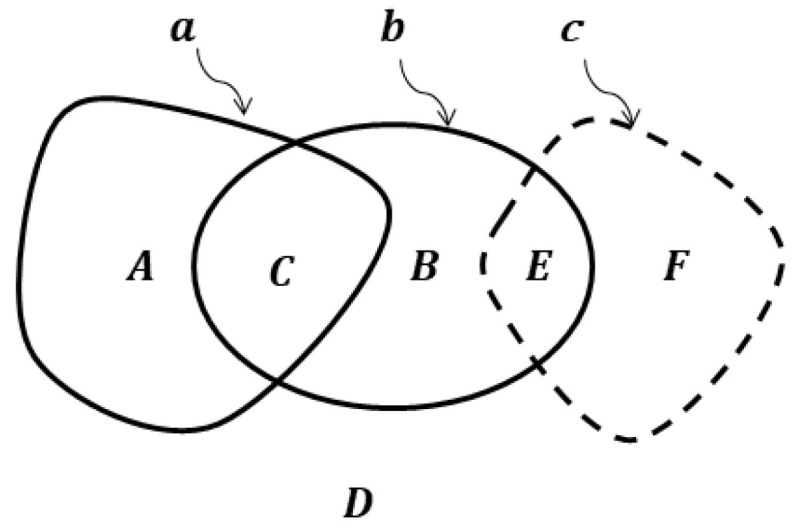
Euler-Venn diagram for ER between figures a, b,and c.

**Figure 3 sensors-23-01247-f003:**
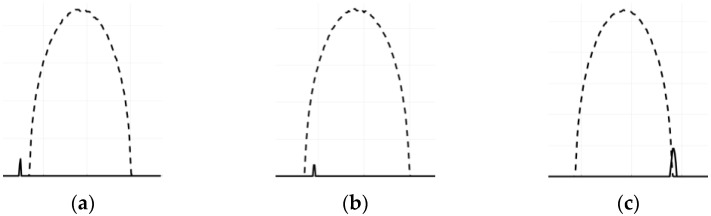
ER between sinograms of an ellipse and a base square. (**a**) Separation; (**b**) Inclusion; (**c**) Intersection.

**Figure 4 sensors-23-01247-f004:**
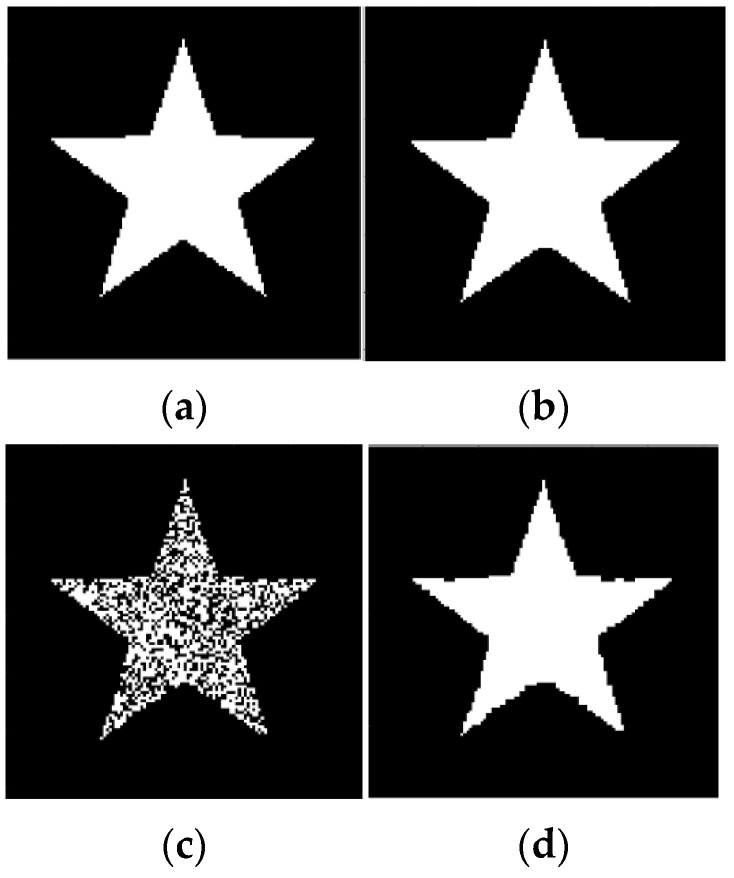
Results of reconstruction of five-pointed star based on qualitative data. (**a**) Original star; (**b**) Reconstructed star; (**c**) Original strong noisy star; (**d**) Reconstructed noisy star.

**Figure 5 sensors-23-01247-f005:**
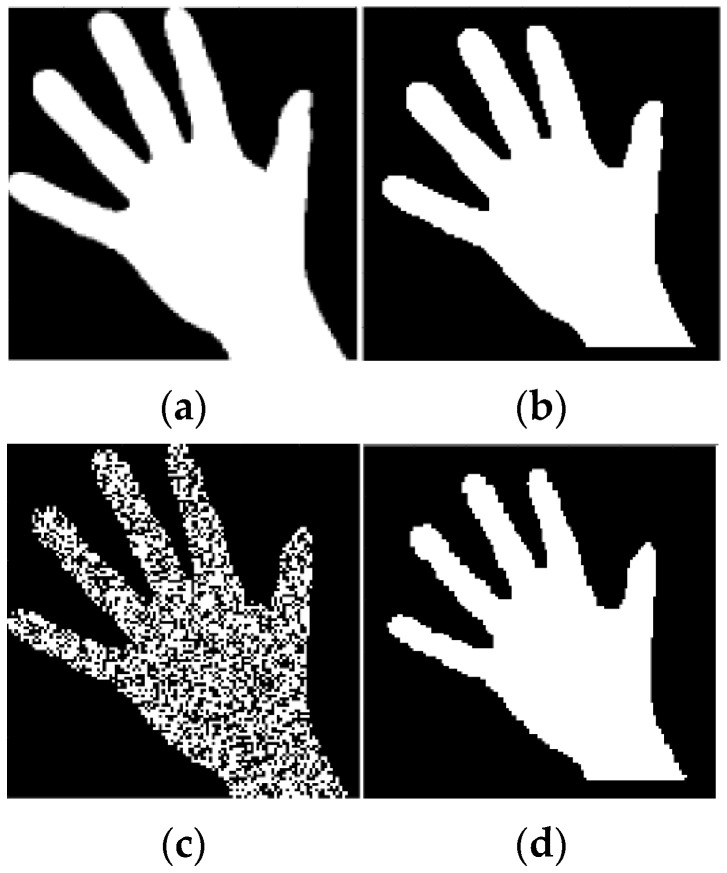
Results of reconstruction of a hand-mask image based on qualitative data. (**a**) Original image; (**b**) Reconstructed image; (**c**) Original strong noisy image; (**d**) Reconstructed noisy image.

**Figure 6 sensors-23-01247-f006:**
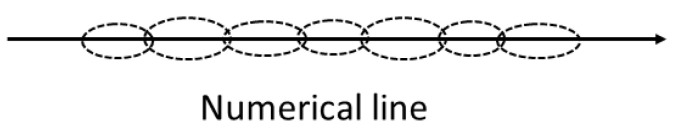
Representation of a numerical line as a chain of intersected spots.

**Figure 7 sensors-23-01247-f007:**
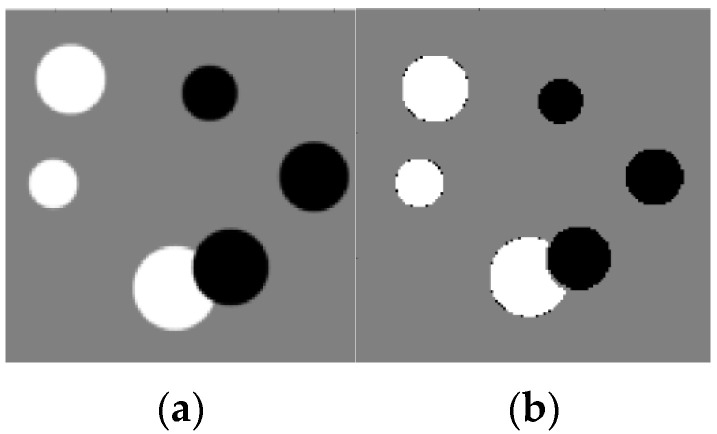
Reconstruction of a circle’s image based on qualitative data. (**a**) Original image; (**b**) Reconstructed image.

**Figure 8 sensors-23-01247-f008:**
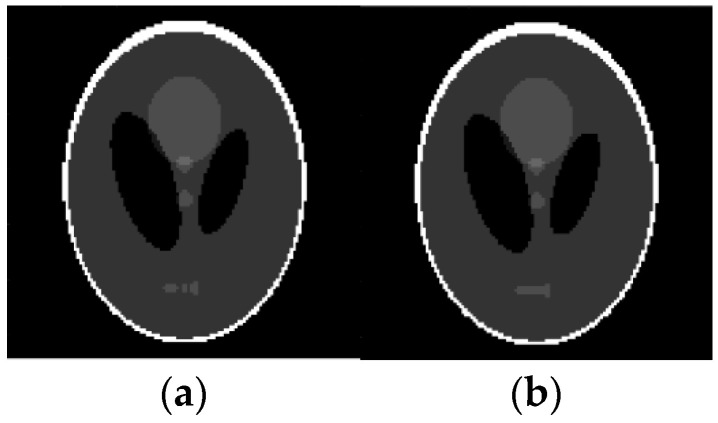
Reconstruction of Shepp-Logan image based on qualitative data. (**a**) Original image; (**b**) Reconstructed image.

**Figure 9 sensors-23-01247-f009:**
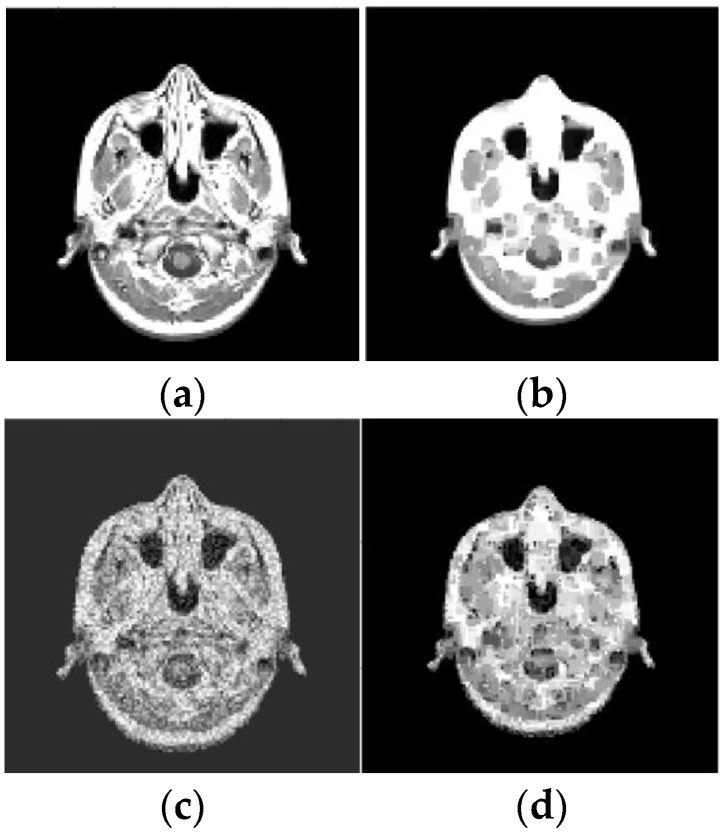
Results of reconstruction of MRI image based on qualitative data. (**a**) Original image; (**b**) Reconstructed image; (**c**) Original noisy image; (**d**) Reconstructed noisy image.

**Figure 10 sensors-23-01247-f010:**
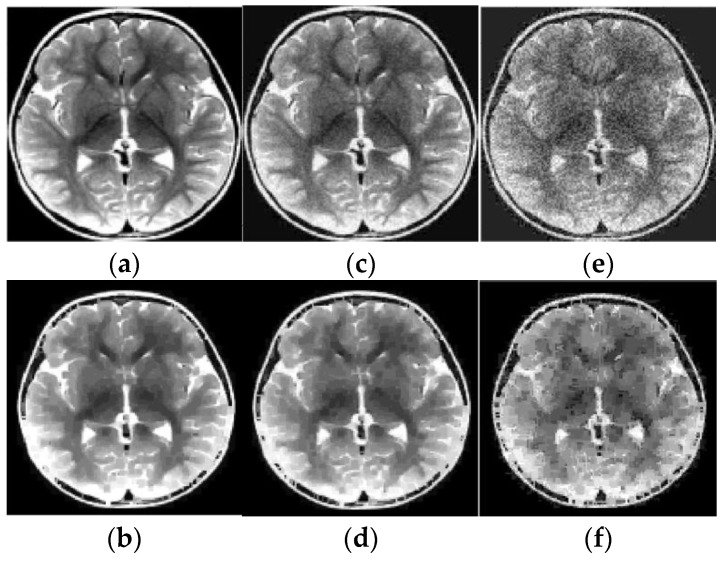
Results of reconstruction of MRI human brain image based on qualitative data. (**a**) Original image; (**b**) Reconstructed image; (**c**) Original low noise image; (**d**) Reconstructed low noise image; (**e**) Original noisy image; (**f**) Reconstructed noisy image.

**Figure 11 sensors-23-01247-f011:**
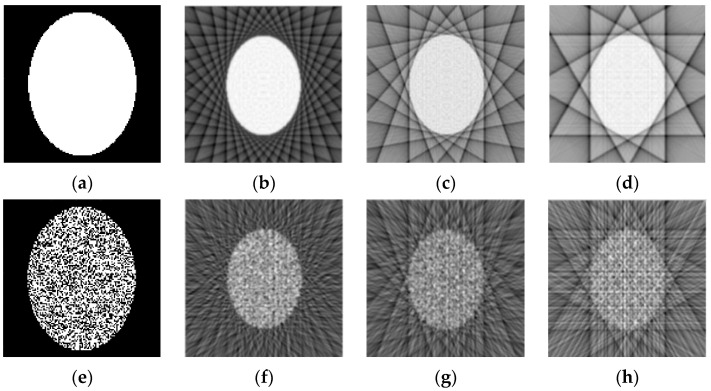
Examples of reconstruction of two ellipses, using their under-sampled parallel-beam sinograms by the back-projection algorithm with Hann filter. (**a**) Original ellipse; (**b**–**d**) Reconstructed images for 18, 9, and 6 projection angles, correspondingly; (**e**) Original strong noisy ellipse; (**f**–**h**) Reconstructed images for 18, 9, and 6 projection angles, correspondingly.

**Figure 12 sensors-23-01247-f012:**
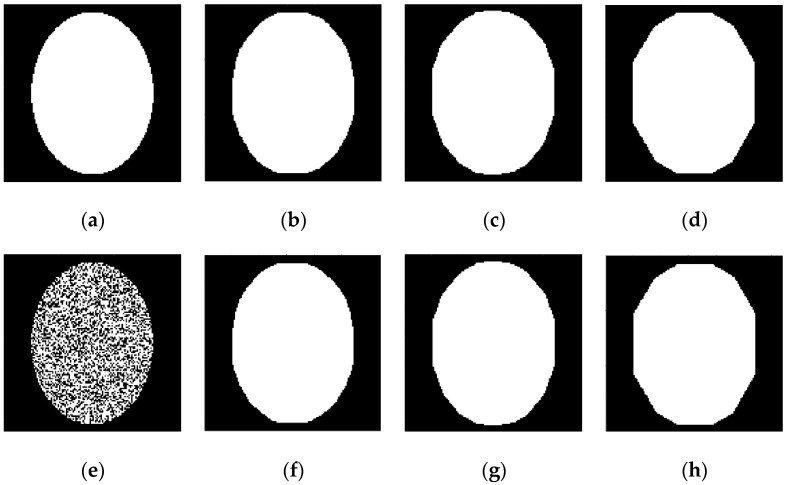
Reconstruction of two ellipses, using their different under-sampled sinograms. (**a**) Original ellipse; (**b**–**d**) Reconstructed images for 18, 9, and 6 projection angles, correspondingly; (**e**) Original strong noisy ellipse; (**f**–**h**) Reconstructed images for 18, 9, and 6 projection angles, correspondingly.

**Table 1 sensors-23-01247-t001:** Some ER of spots.

Elementary Relations	L4 Number
intersection, a><b	$\left[\begin{array}{cc} 1 & 1\\ 1 & 1 \end{array}\right]$
separation, a<>b	$\left[\begin{array}{cc} 0 & 1\\ 1 & 1 \end{array}\right]$
inclusion (more), a>b	$\left[\begin{array}{cc} 1 & 1\\ 0 & 1 \end{array}\right]$
inclusion (less), a<b	$\left[\begin{array}{cc} 1 & 0\\ 1 & 1 \end{array}\right]$
indiscernibility, a≈b	$\left[\begin{array}{cc} 1 & 0\\ 0 & 1 \end{array}\right]$

**Table 2 sensors-23-01247-t002:** Analogies between the concepts of rough sets and spots.

Concepts of the Rough Set Theory	Concepts of the Spots Model
elements of the universe	atomic basis
granules	spots
attributes	basis of spots
attributes values	L4 numbers
boundary region	boundary
lower approximation	inner region
upper approximation	inner region + boundary

## Data Availability

Not applicable.

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
