# Peer review of "Application of the Model of Spots for Inverse Problems"

_sensors, 2023, doi:10.3390/s23031247_

Round 1

Reviewer 1 Report

In this study, the authors proposes application of a new mathematical model of spots for solving inverse problems using a learning method. They also demonstrates application of spot‐based inverse Radon algorithm for binary image reconstruction.

This work is one of the most impressive examples of inverse problems. Expressing inverse problems concretely is very important for spectral theory. Throughout the study, spectral theory and functional analysis are strongly covered. I think that this study will provide a very important motivation to mathematicians working on this subject. 

Author Response

Thank you very much for the positive evaluation of my work and a note about the analogy of my model and the spectral theory. It is interesting to understand the meaning of such an analogy. I developed the spots apparatus using the analogy with the numerical matrix-vector models. Now it makes sense to consider the spectral theory in more detail with a view to the possible application of its ideas to my model.

Reviewer 2 Report

To solve inverse problems, the proposed model uses model of spots as elementary relations between them, and introduces L4 logical numbers that encode these relations. The paper is well organized, and there are three other issues that concern me,

(1) The experimental results should be emphasized in the abstract.

(2) The Quantitative description of the experimental results can be supplemented further.

(3) Since the proposed works well in the image reconstruction, how about the computational efficiency?

Author Response

  • The experimental results should be emphasized in the abstract.

Reply.

Thank you for this comment. I have formulated the experimental results in the abstract as follows:

“The introduced apparatus was verified by solving problems of reconstruction of images, utilizing only qualitative data of its elementary relations with some scanning figures. This article also demonstrates application of spot-based inverse Radon algorithm for binary image reconstruction. In both cases, the spot-based algorithms have demonstrated an effective denoising property.”

  • The Quantitative description of the experimental results can be supplemented further.

Reply.

Thank you for your advice. I added to the text more descriptions for parameters of the simulation. For example, CPU type and RAM size, calculation times, reconstructed misfit errors, noises levels etc.

  • Since the proposed works well in the image reconstruction, how about the computational efficiency?

Reply.

This is a pilot work on an alternative approach to solving inverse problems. Therefore, my goal was to show the principial possibility of using there the spot apparatus, which I think has good prospects. Note that I did not optimize a simple MATLAB program and did not use GPU. The computation times were 6 – 9 minutes. In the future, I intend to work on writing computer programs that are more optimal for possible competition with other learning-based solution methods. In this regard, further research is needed on the application of the developed approach, including for solving the inverse Radon problem for grayscale images, for fan-beam image reconstruction and for inverse problems of electromagnetic scattering.

To increase the efficiency of using the apparatus of L4 numbers, it is possible to write more optimal codes and compile them, parallelize calculations, use GPUs, etc. For the practical application of the proposed algorithms, it is advisable to move to the hardware level of coding operations with L4 numbers and matrices, as well as the creation of specialized processors.

Reviewer 3 Report

Notes:

1. The journal allows the use of a comma without a space when listing references to literary sources, for example, [1,2]. In the text of your article, you adhere to this, with the exception of lines 121, 124.

2. When specifying the dimension of matrices, it is customary to use the symbol “´”, see lines 127, 129, 414, 442, 470, 497, 498, 501.

3. All variables are written in italics, for example, A, B, c, d, r, a, r. See formula (5).

4. Line 268, paragraph indentation may be needed.

5. Line 298. Missing space when listing variables. Must be ?, ?”.

6. Line 312. Should be “(15) – (17)”.

7. Line 382. Should be “figures – squares”.

8. Maybe make Figure 3 brighter and more contrasty. The picture looks faded.

9. Line 524. Should be “objects – spots”.

10. Line 616. Should be “551–596”, то есть без пробелов.

11. Line 689. Should be 833–852.”, то есть без пробелов.

12. Literary references 77, 78 are the same reference. “Wang, Zh.; Zhang, J.; Feng, J.; and Chen, Zh. Knowledge graph embedding by translating on hyperplanes. In Proc. 28th AAAI Conference on Artificial Intelligence, Québec City, Québec, Canada, July 27–31, 2014; pp. 1112–1119.”

Your paper is replete with rather complex formulas. I recommend carefully subtracting the text of the paper in order to eliminate technical errors, maybe I missed something.

I recommend accepting your paper for publication with minimal technical correction.

Author Response

I want to express my gratitude to you for a very careful reading of the text and valuable comments on the errors and omissions found. Here are my responses to your comments.

  1. The journal allows the use of a comma without a space when listing references to literary sources, for example, [1,2]. In the text of your article, you adhere to this, with the exception of lines 121, 124.

Reply. Thank you, I fixed it.

  1. When specifying the dimension of matrices, it is customary to use the symbol “´”, see lines 127, 129, 414, 442, 470, 497, 498, 501.

Reply: Thank you, I changed letter “x” to mathematical symbol of multiplication “×”. In my experience, symbol “×” is also acceptable for the matrix dimension.

  1. All variables are written in italics, for example, ABcdrar. See formula (5).

Reply. Thank you, I fixed it.

  1. Line 268, paragraph indentation may be needed.

Reply. Thank you, I fixed it.

  1. Line 298. Missing space when listing variables. Must be “?, ?”.

Reply. Thank you, I fixed it.

  1. Line 312. Should be “(15) – (17)”.

Reply. Thank you, I fixed it.

  1. Line 382. Should be “figures – squares”.

Reply. Thank you, I fixed it.

  1. Maybe make Figure 3 brighter and more contrasty. The picture looks faded.

Reply. Thank you, I redrawn the images with more line thickness.

  1. Line 524. Should be “objects – spots”.

Reply. Thank you, I fixed it.

  1. Line 616 Should be “551–596”, то есть без пробелов.

Reply. Thank you, I fixed it.

  1. Line 689 Should be “ 833–852.”, то есть без пробелов.

Reply. Thank you, I fixed it.

  1. Literary references 77, 78 are the same reference. “Wang, Zh.; Zhang, J.; Feng, J.; and Chen, Zh. Knowledge graph embedding by translating on hyperplanes. In Proc. 28th AAAI Conference on Artificial Intelligence, Québec City, Québec, Canada, July 27–31, 2014; pp. 1112–1119.”

Reply. Thank you, I fixed it. I mistakenly moved part of the reference to a new line.

Your paper is replete with rather complex formulas. I recommend carefully subtracting the text of the paper in order to eliminate technical errors, maybe I missed something.

Reply. Thank you for your advice. I checked the text and fixed some more errors.

Reviewer 4 Report

A new mathematical model of spots for solving inverse problems is proposed in this paper.

1.In the final experimental results, the noisy and noiseless images are reconstructed, but the effect can only be seen in the pictures. How does this method compare with other learning methods?

2. In the process of reconstruction imaging, can the advantages of the proposed method be compared with other methods in terms of time and calculating power?

3. Can spots model and the concept of L4 vectors and L4 matrix be applied to a wider computing model?

4. There are spelling mistakes in the manuscript, such as“nose‐free” in line 461. Please correct these mistakes.

Author Response

1. In the final experimental results, the noisy and noiseless images are reconstructed, but the effect can only be seen in the pictures. How does this method compare with other learning methods?

Reply. Thank you for your question. I added to the text some more parameters of the reconstructions, including noises levels and reconstructed misfit errors. I think it is difficult to compare my method with other ones at this stage because this is a pilot work on an alternative approach to solving inverse problems. Therefore, my goal was to show the principial possibility of using there the spot apparatus, which I think has good prospects. Note that I did not optimize a simple MATLAB program and did not use GPU.

  1. In the process of reconstruction imaging, can the advantages of the proposed method be compared with other methods in terms of time and calculating power?

Reply. As I said, this is a pilot work on a new approach to solving inverse problems and I did not optimize a simple MATLAB program and did not use GPU. In the future, I intend to work on writing computer programs that are more optimal for possible competition with other learning-based solution methods. To increase their efficiency, it is possible to write more optimal codes for operations with L4 numbers and compile the programs, parallelize calculations, use GPUs, etc. For the practical application of the proposed algorithms, it is advisable to move to the hardware level of coding operations with L4 numbers and matrices, as well as the creation of specialized processors.

  1. Can spots model and the concept of L4 vectors and L4 matrix be applied to a wider computing model?

Reply. Yes, I think my method is applicable for many tasks of solution inverse problems by leaning. For example, I intend to apply the developed apparatus in further research including for solving the inverse Radon problem for grayscale images, fan-beam image reconstruction and for inverse problems of electromagnetic scattering.